# Adjoint Algorithm Design of Selective Mode Reflecting Metastructure for BAL Applications

**DOI:** 10.3390/nano14090787

**Published:** 2024-05-01

**Authors:** Zean Li, Xunyu Zhang, Cheng Qiu, Yingshuai Xu, Zhipeng Zhou, Ziyuan Wei, Yiman Qiao, Yongyi Chen, Yubing Wang, Lei Liang, Yuxin Lei, Yue Song, Peng Jia, Yugang Zeng, Li Qin, Yongqiang Ning, Lijun Wang

**Affiliations:** 1State Key Laboratory of Luminescence and Applications, Changchun Institute of Optics, Fine Mechanics and Physics, Chinese Academy of Sciences, Changchun 130033, Chinazhangxunyu22@mails.ucas.ac.cn (X.Z.); xuyingshuai22@mails.ucas.ac.cn (Y.X.); weiziyuan22@mails.ucas.ac.cn (Z.W.); qiaoyiman23@mails.ucas.ac.cn (Y.Q.); wangyubing@ciomp.ac.cn (Y.W.); liangl@ciomp.ac.cn (L.L.); leiyuxin@ciomp.ac.cn (Y.L.); songyue@ciomp.ac.cn (Y.S.); jiapeng@ciomp.ac.cn (P.J.); zengyg@ciomp.ac.cn (Y.Z.); qinl@ciomp.ac.cn (L.Q.); ningyq@ciomp.ac.cn (Y.N.); wanglj@ciomp.ac.cn (L.W.); 2University of Chinese Academy of Sciences, Beijing 100049, China; 3Xiongan Innovation Institute, Chinese Academy of Sciences, Xiongan 071899, China; 4Jlight Semiconductor Technology Co., Ltd., Changchun 130033, China

**Keywords:** broad-area laser, large scale, mode selection, adjoint algorithm, inverse design

## Abstract

Broad-area lasers (BALs) have found applications in a variety of crucial fields on account of their high output power and high energy transfer efficiency. However, they suffer from poor spatial beam quality due to multi-mode behavior along the waveguide transverse direction. In this paper, we propose a novel metasurface waveguide structure acting as a transverse mode selective back-reflector for BALs. In order to effectively inverse design such a structure, a digital adjoint algorithm is introduced to adapt the considerably large design area and the high degree of freedom. As a proof of the concept, a device structure with a design area of 40 × 20 μm^2^ is investigated. The simulation results exhibit high fundamental mode reflection (above 90%), while higher-order transverse mode reflections are suppressed below 0.2%. This is, to our knowledge, the largest device structure designed based on the inverse method. We exploited such a device and the method and further investigated the device’s robustness and feasibility of the inverse method. The results are elaborately discussed.

## 1. Introduction

Broad-area lasers (BALs) are high-power devices that play a pivotal role in industrial processing [1], medical equipment [2], optical communications [3], photonic pumping sources [4], laser printing [5], and many other domains [6]. A notable feature of BALs is their emission region width, which significantly exceeds that of conventional lasers. The sizeable lateral dimension of BALs enables the use of a higher drive current without damaging the device’s integrity, which produces a higher-power laser output. However, as the lateral width increases, the beam quality of such lasers deteriorates severely.

Initially, the filamentation effect, stemming from carrier-induced anti-guiding, was considered to be the main culprit behind the degradation of beam quality at elevated injection levels during the early phases of BAL research. However, the contemporary research community now leans towards recognizing the complex multi-mode behavior observed in the transverse direction as the predominant mechanism responsible for this deterioration. The presence of high-order lateral optical modes, discernible even at very high current injection levels through high-resolution spectral near-field and far-field measurements, underscores the critical need for strategies that enable spatial mode selection and filtering [7]

At present, the cutting-edge approaches for spatial mode selection and filtering in BALs encompass the use of digital planar holograms [8], photonic crystal structures [9], loss tailoring [10], external feedback for beam shaping [11], and evanescent coupling [12]. While these methods have achieved significant success in enhancing beam quality, they seldom enable high-power BAL beams to reach near the diffraction limit.

The challenges faced by current techniques in achieving exceptional performance in BAL devices can be attributed to several factors. These include the ineffectiveness of mode filtering, inaccuracies in mode selection, and a lack of robustness under demanding, high-level injection conditions. For example, techniques such as evanescent coupling rely on the proximity of the waveguide to a resonant structure to selectively couple out high-order modes. However, as the width of the waveguide increases, the effectiveness of this coupling decreases significantly, leading to a reduced ability to filter out undesired modes. Spatial mode filters, including photonic crystals and digital planar holograms, are designed to differentiate between the fundamental mode and high-order modes. They often operate based on the gross differences in the modes’ characteristics. However, these filters may not provide the fine-tuned discrimination needed to accurately select the desired mode, especially in complex scenarios with multiple closely spaced modes. The loss tailoring method adjusts the loss distribution within the BAL device to shape the beam profile. However, this method is highly sensitive to the operating conditions of the device. At very high current injections, the non-linearities in the device’s behavior become more pronounced, making it difficult for the loss tailoring method to maintain effective control over the beam quality. To sum up, the theoretical models and physical intuitions that guide the design of selective filters may not fully capture the complex dynamics of BAL devices, particularly under extreme operating conditions. This gap between theory and practice makes it challenging to design a filter that is simultaneously effective, accurate, and robust.

On the other hand, inverse design techniques and algorithms have rapidly evolved in recent years. The metastructure waveguide in inverse engineering offers high flexibility and customizability, allowing for precise modulation of optical properties according to requirements. Moreover, the latest review offers an extensive collection of research focusing on the inverse design of photonic devices [13]. Inverse design has surpassed the optical capabilities conventionally achieved through intuitive design methods while markedly simplifying the design process. This technique has led to the development of advanced photonic devices, such as polarization beam splitters [14], mode converters [15], mode demultiplexers [16], power splitters [17], logic gates [18], and nonlinear optical devices [19]. The exceptional capability to design photonic devices that outperform their conventional counterparts could potentially unlock the development of the aforementioned on-demand selective spatial mode filtering structures, offering significant advancements in both photonic and semiconductor laser technologies.

In this paper, we introduce a novel selective spatial mode filtering reflector structure complemented by its inverse design strategy. This design aims to optimize high-power broad-area laser devices, enabling them to produce outputs that closely approach the diffraction limit. The strategy focuses on identifying structures that can precisely select the fundamental lateral optical mode across the transverse direction of the BAL waveguide, significantly boosting its reflectivity. It is also tailored to effectively diminish the reflectivity of all other higher-order lateral optical modes. This reflector structure, positioned externally to the active region and functioning as part of the laser’s cold cavity, is anticipated to be less susceptible to variations in the injection level of BALs, thereby enhancing its robustness. Given that the design area of the proposed structure spans a significantly larger footprint (around 1000 μm^2^) compared to the typical dimensions handled by conventional inverse design methods for photonic devices, which usually operate on scales of tens of square micrometers, we employ a digital adjoint sensitivity analysis to mitigate the computational resource constraints. The selection of the Figure of Merit (*FoM*) and the formulation for the reflectivity of lateral optical modes have been meticulously optimized. After an extensive optimization process of approximately five days, the structure achieved a remarkable reflectivity of 90% for the fundamental mode. Meanwhile, it significantly reduced the reflectivity of the first through fourth higher-order modes to below 0.2%. This accomplishment renders the structure highly appropriate for use as a reflective cavity mirror in laser applications. Our work not only serves as a proof of concept but also shows the potential for developing large-scale photonic devices with an unprecedented level of design flexibility.

## 2. Design Principle and Optimization Strategy

The objective of our design is to identify device structures within a defined region that serve dual roles: acting as a highly reflective rear mirror for the cavity of a broad-area semiconductor laser and functioning as a lateral mode selective filter. The latter role involves enhancing the reflectivity for the fundamental mode while concurrently diminishing the reflection for all higher-order lateral optical modes. A schematic representation of the intended functionality with a possible laser device setup is depicted in Figure 1.

Given the absence of a theoretical framework to guide the discovery of a reflector that can deliver tailored performance with the requisite levels of efficiency and precision, it becomes prudent to consider employing an optimization-centric inverse design methodology. This approach is well-suited to tackle the nature of this challenge, often referred to as a “black box” problem, by working backwards from the desired outcomes to engineer a solution that meets the specified performance benchmarks.

To fully harness the potential of this sophisticated strategy, meticulous adherence to the prescribed design criteria is paramount. Prior to initiating the iterative optimization process, it is essential to contemplate a multitude of factors. These considerations ensure that the approach is not only initiated correctly but also tailored to achieve the most effective and efficient outcomes.

### 2.1. Definition of the Performance Criteria

The initial phase of this endeavor hinges on the explicit delineation of the reflector’s objectives. As previously indicated, the efficacy of the target structure is primarily gauged by the reflectivity of the eigen propagation modes, which are characterized by diverse field distributions throughout the transverse plane of the wide waveguide. With this in mind, it becomes essential to thoroughly analyze the expression of reflectivity associated with each lateral optical mode and to derive these values through rigorous simulation analysis. This foundational work is paramount for capturing the nuanced behavior of the reflector and for informing the subsequent stages of design and optimization.

To begin with, the field distribution of optical lateral modes, which are permitted to propagate within the designated BAL waveguide at the target frequency, is computed. Subsequently, a source is stimulated using the calculated field distribution of a specific lateral mode, enabling the propagation of the electromagnetic wave within the intended waveguide. Once the wave has traversed the predefined region, a monitoring plane is established to assess the amplitude and distribution of the reflected field within the waveguide. The reflectivity for each mode is quantified by the correlation coefficient between the reflected field and the corresponding lateral mode. For instance, in the case of a transverse electric (TE) polarized field, the formula is as follows:(1)rTEi2=14|∫S(E(x′)×Hi*(x′)+Ei*(x′)×H(x′))·dS|2|∫SRe(Ei(x′)×Hi*(x′))·dS|2
where *rTE_i_*^2^ denotes the reflectivity of the *i*-th order *TE* mode; *S* is the cross-section on the port of the waveguide; *E_i_* and *H_i_* correspond to the norm electronic and magnetic fields of the *i*-th order mode, respectively; and *E* and *H* denote the electric and magnetic field distributions monitored at the set plane.

Following the derivation of the reflectivity for each optical lateral mode through corresponding simulations, it is essential to establish an *FoM* to assess the overall performance. In this study, we introduce a general expression for the *FoM* function as outlined below:(2)$$F=FoM=C_{0}\cdot {rTE_{0}}^{2}-C_{1}\cdot {rTE_{1}}^{2}-C_{2}\cdot {rTE_{2}}^{2}-\ldots -C_{i}\cdot {rTE_{i}}^{2}$$

In this formulation, *C*_0_ is designated as the coefficient for the fundamental lateral mode. The coefficients *C*_1_ to *C_i_* correspond to the high-order lateral modes that are permissible. These coefficients are adjustable parameters within the model. For our standard cases, we have set *C*_0_ to 1, and *C*_1_ to *C_i_* are proportional to 1/N, where N represents the total count of high-order lateral modes. A mini discussion on the selection of these coefficients is provided in the “Results and Discussion” section.

### 2.2. Modeling and Simulation Methods

It is widely recognized that the simulation and design of a three-dimensional photonic structure is computationally expansive. The situation becomes even worse when the design method is iterative and the design region is considerably large, both of which, unfortunately, happen to be true in our case. To mitigate the computational constraints and ensure that the design process remains feasible within a reasonable timeframe, it becomes imperative to employ two-dimensional simulations that offer sufficient accuracy to closely emulate the behavior of the 3D model. This approach allows for a more efficient use of computational resources while still achieving the desired level of precision in the design and analysis of the photonic structure.

In general, simulations of BAL structures often involve the propagation of light waves in specific lateral modes, which are subsequently subject to perturbations caused by variations in the refractive index of the materials within the waveguide. To accurately capture the behavior of these 3D models, the 2D Effective Index Method (EIM) can be employed under certain constraints regarding the refractive index contrast. This method simplifies the complex 3D structure into a 2D cross-section, allowing for a more tractable analysis while still maintaining a high degree of accuracy.

The EIM is particularly useful in scenarios where the refractive index variations are confined within a plane and the structure exhibits translational symmetry in the direction perpendicular to this plane. By approximating the 3D waveguide with a 2D effective index, the computational complexity is significantly reduced without sacrificing the precision required for the analysis [20].

Recent studies have demonstrated the effectiveness of the EIM with high precision, especially in the context of photonic inverse design applications [21]. Inverse design involves the optimization of a photonic structure’s parameters to achieve a desired response, such as a specific mode profile. The EIM’s ability to closely approximate the behavior of 3D photonic structures makes it a valuable tool in this process, enabling the efficient and accurate optimization of complex photonic devices.

In order to effectively implement the EIM in our simulations, we first calculate the effective indices for the slab waveguide mode in both regions where the refractive index profile remains constant and where it varies. These calculated effective indices are then utilized to represent the material’s refractive index in a 2D configuration, simplifying the simulation while maintaining a level of accuracy that is adequate for our purposes.

Moreover, to ensure the sufficient accuracy of the EIM approximation, an additional constraint must be established. The primary source of error in this method stems from the scattering and excitation of higher-order slab modes. Consequently, the precision of the 2D model is contingent upon the strength of the coupling to these additional scattering channels. To address this, a straightforward approach is to confine the difference between the effective indices of interest within a narrowly defined range. By this means, we can achieve a high degree of precision in our 2D approximation.

This strategy ensures that the EIM provides a reliable representation of the 3D waveguide behavior, particularly in terms of mode confinement and propagation characteristics. By carefully controlling the parameters and constraints within the EIM framework, we can effectively approximate the complex interactions of light within the photonic structure, thereby enabling accurate and efficient simulation of the waveguide’s performance.

Upon the adoption of the EIM approximation, we proceed to utilize 2D Time-Domain Finite Difference (FDTD) analysis as our primary simulation tool throughout the iterative optimization process. Subsequently, to validate the final design structures, the Finite Element Method (FEM) in the frequency domain is also employed. This multifaceted approach combines the efficiency of EIM with the powerful analysis capabilities of FDTD and FEM, thereby yielding a comprehensive and rigorous design optimization workflow.

### 2.3. Inverse Algorithm and Optimization Strategy

While the 2D approximation method has effectively mitigated the computational intensity of electromagnetic (EM) simulations, the development of structures with large dimensions and an extremely high degree of freedom (DOF) continues to present a formidable challenge. Direct binary search algorithms, which have computational requirements directly proportional to the number of pixels, can have daunting timelines. For instance, in our case, optimizing a structure with 80,000 DOF could entail a single iteration taking nearly a hundred days, with no certainty of achieving a satisfactory solution, making this approach impractical.

Heuristic algorithms, despite offering increased flexibility, do not necessarily offer a reduction in computational demand and, in some instances, may even increase it. In the face of such challenges, the critical element is the discovery of design algorithms that are both efficient and effective.

The adjoint method emerges as a breakthrough, transcending the efficiency limitations characteristic of traditional optimization strategies. Anchored in the principle of electromagnetic field reciprocity, the adjoint method facilitates the acquisition of gradient information across the entire design space with just a forward and an adjoint simulation. This capability is pivotal, as it underpins the theoretical feasibility of an efficient inverse design methodology. By harnessing the adjoint method, it becomes feasible to significantly streamline the optimization process, enabling the exploration and realization of complex photonic structures with large DOFs that were previously considered computationally intractable [22].

Assisted with Figure 2, we demonstrate the workflow and strategy of inverse design based on the adjoint algorithm as follows.

After configuring the foundational parameters, such as the operating wavelength, the background refractive index, and the BAL waveguide width, the effective indices for both perturbed and unperturbed areas are calculated using the 2D EIM approximation before the simulation commences.

The iterative procedure is structured as follows:(1)Calculate the field distribution for the fundamental lateral mode within the BAL waveguide.(2)Introduce a source with the calculated field distribution and allow it to propagate through the design region using FDTD simulation.(3)Set up a monitoring plane on the reflecting side to capture the reflected field and use Equation (1) to compute the mode’s reflectivity.(4)Concurrently, record the electric field distribution in the design region, denoted as *E^Fwd^*.(5)Stimulate a source at the monitoring plane and let the wave propagate back through the design region.(6)Record the electric field in the design region, referred to as *E^Adj^*.(7)Compute the gradient of the fundamental mode with respect to changes in the refractive index within the design region using the following formula:
$$\frac{\partial rT{E_{0}}^{2}(x,y)}{\partial \varepsilon_{r}}=\varepsilon_{0}Vℜe\left[{E_{0}}^{Fwd}(x,y)\cdot {E_{0}}^{Adj}(x,y)\right]$$

(8)Repeat steps 1 to 7 for all permissible lateral modes.(9)Utilize Equation (2) to amalgamate all the partial derivative terms to formulate the *FoM* function.(10)Calculate the average of the gradients at all grid positions within each pixel.(11)Identify the point with the minimum gradient value on the *FoM* function plane and flip its refractive index.(12)Update the real-time loss, *FoM*, and reflectivity for all viable lateral modes.(13)Continue this iterative process until the loss falls below an acceptable threshold.

This systematic approach ensures a thorough optimization of the waveguide design, taking into account the behavior of multiple lateral modes and their respective contributions to the overall performance. By iteratively refining the design based on the calculated gradients and the *FoM*, the process converges towards a solution that minimizes loss while maintaining the desired properties of each mode.

As a demonstration of the effectiveness of the inverse design strategy outlined previously, a reflector structure for a BAL waveguide with a width of 20 μm has been designed. The key parameters for this design are compiled in the Table 1.

This proof of concept showcases the application of the inverse design strategy in creating a BAL waveguide reflector structure tailored to specific performance criteria. The detailed parameters provided in the table are crucial for understanding the design constraints and objectives, ensuring that the final structure meets the desired operational specifications. By following the iterative optimization process, the design evolves to achieve the target reflectivity with an acceptable level of loss, demonstrating the practicality and robustness of the inverse design approach in photonic device engineering.

## 3. Results and Discussions

### 3.1. Benchmark Result

Following an extensive optimization process involving 4000 iterations, the *FoM* achieved a high value of 90.00%. This result is accompanied by a reflectance of 90.04% for the TE_0_ mode, which is the fundamental lateral mode of interest. This high reflectance indicates that the designed reflector structure is highly efficient in confining and reflecting the light within the TE_0_ mode.

Moreover, the optimization process successfully suppressed the reflectivity of all higher-order modes to below 0.2%. This suppression is crucial for ensuring that high-power BAL operation in single lateral mode with excellent beam quality.

The effectiveness of the optimization is visually demonstrated in Figure 3, which indicates the improvement in *FoM* and the suppression of higher-order modes over the course of the optimization iterations.

Meanwhile, the structural details of the final optimized design are depicted in Figure 4. This figure provides a visual representation of the waveguide’s 3D profile, highlighting the variations in refractive index that contribute to the desired optical performance.

The successful outcome of this optimization process underscores the power and precision of the inverse design methodology, particularly when combined with the EIM and the adjoint method. It showcases the capability of these methods to yield high-performance photonic structures that meet stringent design specifications, paving the way for the development of advanced photonic devices with controlled and predictable optical properties.

More specifically, we embarked on a comprehensive series of comparative studies, meticulously analyzing the impact of various design elements on the performance of photonic systems. To set a control, the simulation parameter configuration and the optimized result that have been described in Section 2 were used as a benchmark for parallel comparison.

### 3.2. Universality of Inverse Method for BAL Design

The device developed through the inverse engineering approach is characterized by its design universality, which is pivotal for its adaptability to a range of geometric dimensions. This universality is critical for aligning with the requirements of high-power BAL devices in a variety of application scenarios.

In Figure 5, the photonic performance of structures with different footprints is showcased, with each structure having undergone the same number of optimization iterations. The uniformity in the optimization process across various sizes ensures that each structure achieves a high level of performance, regardless of its physical scale.

Figure 5c,f highlight the robustness of the optimization process. These figures demonstrate that the outcomes are remarkably insensitive to changes in the device’s physical scale, a testament to the reliability and consistency of the optimization algorithm. This insensitivity to scale is a key attribute, as it ensures that the photonic structures will perform consistently across a wide range of sizes, thereby expanding the potential application field for these devices.

The ability to maintain high performance across different scales is particularly beneficial in the field of photonics, where devices may need to be integrated into various systems with different space constraints or may be required to operate under diverse conditions. The robustness and adaptability of the design ensure that these photonic structures can be effectively utilized in a multitude of applications, from high-density optical interconnects to scalable optical networks and beyond. This versatility makes the inverse engineering approach not only a powerful design tool but also a valuable asset in the ongoing development and innovation of photonic technologies.

### 3.3. Optimization of FoM Functions

Identifying an effective *FoM* to serve as the optimization target, which could lead to accelerated convergence rates and improved optimization outcomes, presents a promising avenue for future exploration in the field of photonics inverse design. For this reason, we defined a more general form of *FoM*:$$FoM={rTE_{0}}^{\alpha }-k\cdot ({rTE_{1}}^{\alpha }+{rTE_{2}}^{\alpha }+{rTE_{3}}^{\alpha }+{rTE_{4}}^{\alpha })$$
where *k* is the attenuation coefficient for higher-order modes and α is the exponential coefficient for the modes’ electric field reflectivity. For the benchmark, *k* = 0.25 and *α* = 2. Figure 6 depicts the impact of parameter configurations for different *FoM*s on the optimization process. As shown in Figure 6c, with the increase in *k*, the convergence rate decreases, and the oscillations in the convergence curve become more pronounced. This can be attributed to the fact that as the weight coefficient of higher-order modes increases, the constraints become more stringent, significantly affecting the convergence characteristics. Regarding *α*, it is apparent that an *α* value of 2 results in enhanced optimization performance. Moreover, in physical terms, this corresponds to the reflectivity of the mode’s energy, rendering such an *FoM* more intuitive and straightforward to interpret. Considering the simulation results from Section 2, we believe that setting k to 0.25 and *α* to 2 is the rational choice, at least for our particular task.

### 3.4. The Choice of Pixel Size

The prevailing photonic platforms predominantly utilize 193 nm photolithography technology, which aligns with a minimum achievable feature size of about 90 to 180 nm. Here, we designed structures with pixel sizes of 100 nm, 200 nm, and 400 nm, and their optimization processes, as well as their final optimized layouts, are illustrated in Figure 7. Larger feature sizes imply easier manufacturability, but, on the other hand, they restrict the design space, leading to less satisfactory device performance, which can be verified in Figure 7c. Although the model with a 200 nm pixel size did not demonstrate superior optical performance in simulations compared to the 100 nm feature size model, the fabricated device with 200 nm features might closely align with its simulated results. However, there may be a significant disparity between the experimental and simulation outcomes for the device with 100 nm features. Therefore, the pixel size of 100~200 nm is worth researching and manufacturing. As for the larger size, the design space is greatly sacrificed and there is no more research value.

### 3.5. Structure Optimization with Lower Refractive Index Contrast

Although the parameter configuration described in the second section is capable of achieving commendable design capabilities, it requires a substantial refractive index contrast (Δ*n* = 1.36) between the etched and non-etched regions. This may necessitate a very deep etching depth, making its implementation in the manufacturing process challenging. To alleviate this problem, we investigated the feasibility of achieving comparable optical performance by adopting a lower refractive index difference, with the simulation results presented in Figure 8. As the reflective index difference decreases, there is a noticeable slowdown in the convergence rate. When the refractive index difference is 0.4, the *FoM* is still rising and has not yet converged to a stable value, which may require more optimization time. Based on the results presented in Figure 8c, if achieving a refractive index parameter of 1.36 proves difficult in fabrication, a refractive index difference of 0.68 is a viable compromise.

### 3.6. Operating Bandwidth Simulation

Additionally, we conducted simulations of the operation bandwidth of the device, as depicted in Figure 9. Although the device’s bandwidth is limited to merely 2 nm (with the *FoM* greater than 0.8 within the range of 1449 to 1551 nm), this characteristic is advantageous for applications in laser systems, since a narrower bandwidth contributes to a more precise single-frequency attribute in lasers.

### 3.7. Robustness Analysis

In the fabrication process of photonics devices, over-etching and under-etching are two common phenomena. To address this, we conducted pre-simulations of varying degrees of under-etching and over-etching and analyzed their significant impact on the optical properties of the device, as shown in Figure 10. The simulation result shows that the structure possesses an etching tolerance of approximately ±2.5 nm for both under-etching and over-etching. Although the robustness in this aspect is unsatisfactory, we aim to initiate further exploration. We hope this work will encourage a more comprehensive consideration of etching robustness in future research.

### 3.8. Potential of Extending the Width of the Reflector

In the cases we have demonstrated, the width of the target BAL waveguide, along with the designed reflector structure, is around 20 μm. This dimension is notably smaller than the widths typically found in standard BAL devices, which generally range from 80 μm to 400 μm. However, the areas of the design regions in our cases are substantial, nearing 1000 square micrometers, which, to our knowledge, represents one of the largest areas reported in photonic inverse design.

The primary factors influencing this design choice are the availability of computational resources and the necessity of maintaining an acceptable timeframe for the design process. In our demonstrated cases, each design process was completed in approximately 122 h using a high-performance computing platform equipped with an i9-13900k CPU and 128 GB of RAM. As the width of the BAL increases, the computational time required for the design process is expected to increase dramatically.

Three main factors are believed to be the primary influencers of the inverse design procedure as the width of the BAL extends:**a.** **Complexity of the Simulation:** With an increase in the width of the BAL, the simulation must account for a larger number of modes and a more complex electromagnetic field distribution. This leads to a higher computational demand as the solver must process additional data and perform more iterations to reach a convergent solution.**b.** **Size of the Design Space**: A wider BAL waveguide increases the design space, which means more variables need to be optimized. This results in a larger search space for the optimization algorithm to navigate, requiring more computational resources to explore and fine-tune the design parameters.**c.** **Size of the Simulation Area:** With the increase in the BAL width, the area of the design region must correspondingly increase to accommodate the functional necessities of more intricate scenarios. Consequently, simulations must encompass a significantly larger area to meet these requirements. The increase in the design region’s area directly impacts the simulation process. A larger area means that the electromagnetic solver must handle a greater number of grid points to accurately model the wave propagation and interactions within the waveguide. This, in turn, leads to an increase in the computational resources needed, such as memory and processing power, and extends the time required for each simulation run.

Taking these factors into account, the required computational resources and the resulting time consumption are anticipated to scale up with the fourth power of the increase in width. This relationship is rooted in the fact that as the width of the BAL waveguide increases, the two-dimensional area of the simulation domain also grows, as well as the number of viable lateral modes and design degrees of freedom.

This overly simplified approximation that the computational resources and time consumption scale up with the fourth power of the increase in the width of the BAL waveguide suggests a significant increase in the time required for design as the width grows. According to this approximation, if a design with a width of 25 μm takes approximately 122 h to complete, then the time required for a design with a width of 50 μm would be 82 days. For a design with a width over 100 μm, the time would be even more extensive, around 7 years, assuming no optimizations or efficiency improvements.

While this approximation provides a rough estimate, it is important to note that it does not account for potential optimizations, advancements in computational methods, or the use of parallel processing, which could mitigate the impact of this scaling. Nevertheless, in practice, the actual time required would likely be much less due to factors such as the following:**a.** **Algorithm Optimizations:** Although the adjoint algorithm is assumed to be a very efficient method, recent studies have suggested a great potential for further improvement of this algorithm. Improved algorithms can reduce the number of required iterations as well as increase the efficiency of the simulation process.**b.** **Parallel Computing:** Utilizing parallel processing can distribute the computational load across multiple processors or computing nodes, significantly reducing the overall time.**c.** **Hardware Advances:** Advances in computing hardware, such as faster CPUs and GPUs, can accelerate the simulation and optimization processes.**d.** **Adaptive Meshing:** Using adaptive meshing techniques can refine the simulation grid only where it is needed, reducing the overall number of grid points.**e.** **The Size of the Design Region:** The assumption of a linear increase in the length of the design region corresponding to the increase in the BAL width may not accurately reflect the actual behavior of devices. As discussed in Section 3.2, the decrease in the length of the design region does not lead to a significant drop in device performance. This observation suggests that there is room for further optimization of the design parameters and that the increase in device length may not be as critical as initially thought.

To sum up, the selection of BAL waveguide widths and the corresponding design of reflector structures described in this study are primarily guided by the availability of reasonable computational resources and the need to maintain an acceptable timeframe for the design process. While it may initially seem daunting to design a reflector for a typical BAL with a width exceeding 100 μm using the current computational setup, the outlook remains optimistic.

The potential for optimization is significant, and as research progresses, there is an expectation that advancements in both software and hardware will play a crucial role in enhancing the design process. Improved algorithms, more efficient use of computational resources, and the development of high-performance computing platforms can all contribute to reducing the time and computational demands associated with the design of larger BAL waveguides.

Moreover, innovative techniques such as adaptive meshing, multiscale modeling, and sophisticated optimization algorithms can lead to more efficient simulations and a faster turnaround time for the design optimization process. These methodological improvements, coupled with the continuous enhancement of computational capabilities, open up the possibility of scaling up the width of BAL waveguides to more practical sizes without compromising performance or feasibility.

In essence, while the design of on-demand structures with larger dimensions presents challenges, the ongoing progress in computational technologies and optimization strategies offers a promising path forward. This progress is expected to enable the design and fabrication of high-power laser devices that meet the demands of current and future applications, pushing the boundaries of what is achievable in the field of semiconductor lasers.

## 4. Conclusions and Future Outlook

In this paper, we have successfully harnessed the power of gradient-based optimization strategies underpinned by the adjoint algorithm to develop a transverse mode selective structure tailored for high-power laser applications. By carefully analyzing the technological challenges facing current solutions and meticulously choosing the *FoM* functions, the proposed method was able to demonstrate a design of reflector structure that has achieved remarkable results on a 40 × 20 μm^2^ optimization area, with over 90% reflectivity for the TE_0_ mode and effectively suppressing higher-order modes to less than 0.2%.

Our demonstration serves as a pioneering example, showcasing the adaptability and flexibility of the inverse design approach with a significantly large design area. This innovative strategy marks a first in the realm of high-power laser design, breaking new ground in the application of advanced optimization methodologies.

By demonstrating the potential of this optimization methodology, we encourage others to explore its application to different material systems, structures, and functionalities. The adaptability of our approach holds promise for the advancement of both photonic technologies and semiconductor laser technologies, paving the way for the development of new devices with enhanced performance and efficiency.

## Figures and Tables

**Figure 1 nanomaterials-14-00787-f001:**
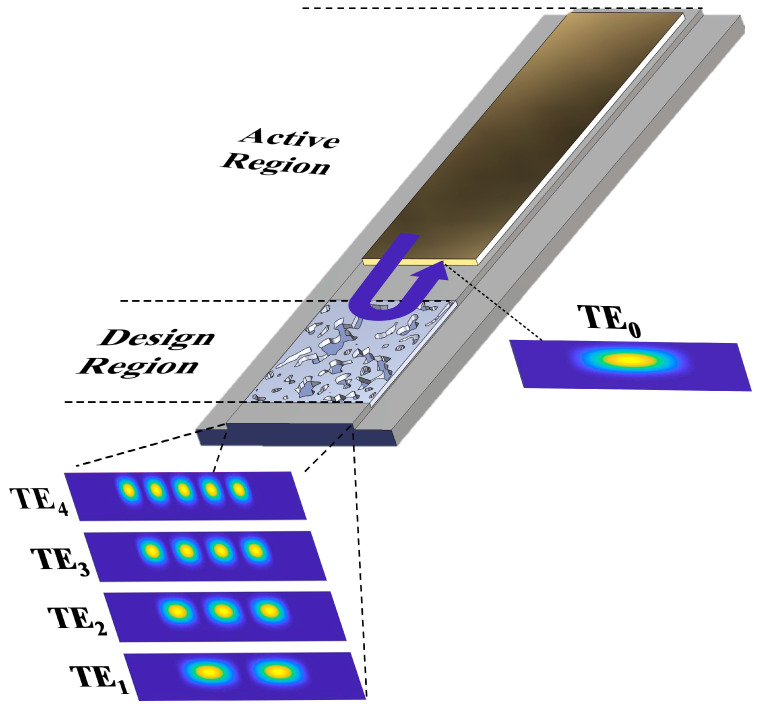
Schematic representation of transverse mode selective reflector in BAL.

**Figure 2 nanomaterials-14-00787-f002:**
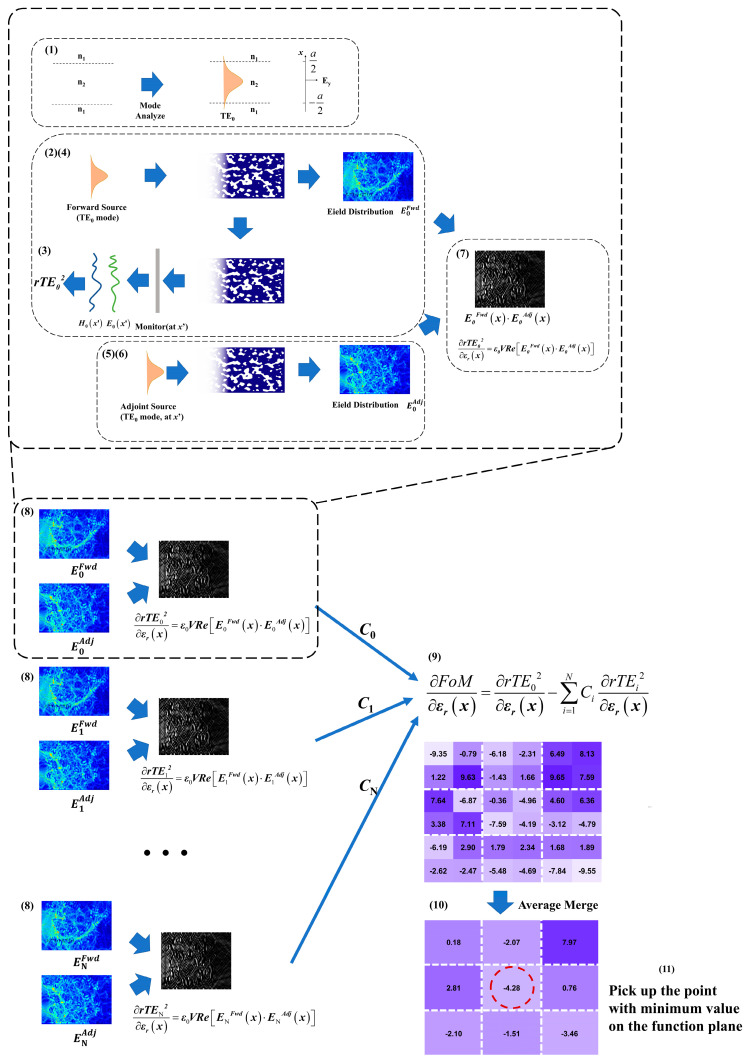
Schematic diagram of adjoint algorithm-based inverse design workflow for selective mode reflecting metastructure (**1**–**11**).

**Figure 3 nanomaterials-14-00787-f003:**
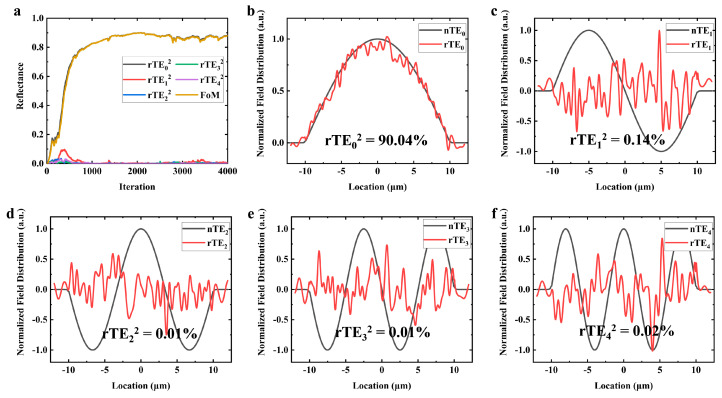
Device performance optimized by the adjoint algorithm. (**a**) *FoM* and reflectivity for each order mode evolution during the optimization. The *FoM* reaching its peak value at the 2065th iteration of optimization. (**b**–**f**) The standard electric field of each order mode (black lines) and their reflected fields (red lines).

**Figure 4 nanomaterials-14-00787-f004:**
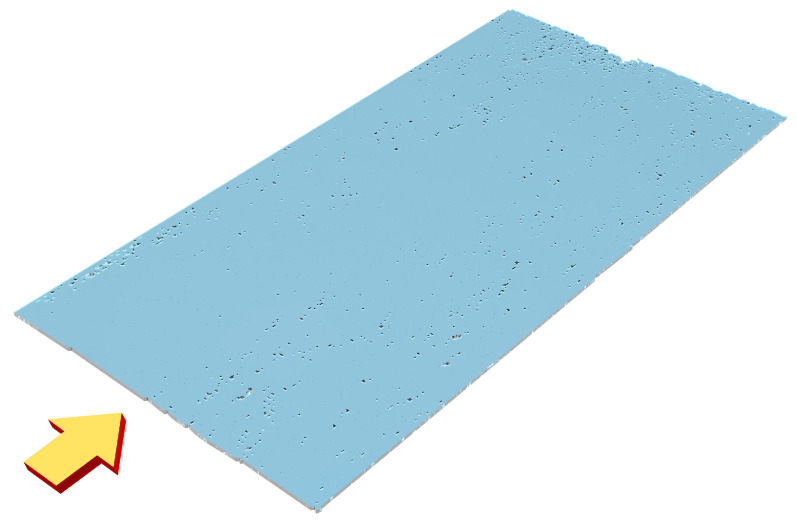
The optimized 3D profile of the metasurface waveguide structure. The direction indicate by the yellow arrow is the direction of the incident light.

**Figure 5 nanomaterials-14-00787-f005:**
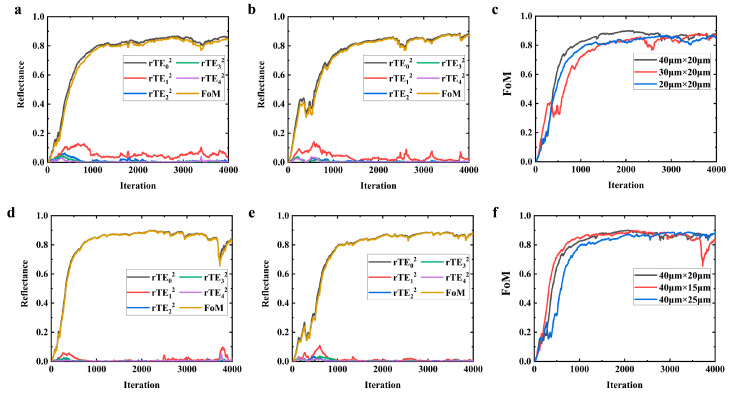
Comparison of optimization processes and outcomes for structures with varied geometry sizes: (**a**) 20 μm × 20 μm; (**b**) 30 μm × 20 μm; (**c**) comparison with benchmark (40 μm × 20 μm); (**d**) 40 μm × 15 μm; (**e**) 40 μm × 25 μm; (**f**) comparison with benchmark (40 μm × 20 μm).

**Figure 6 nanomaterials-14-00787-f006:**
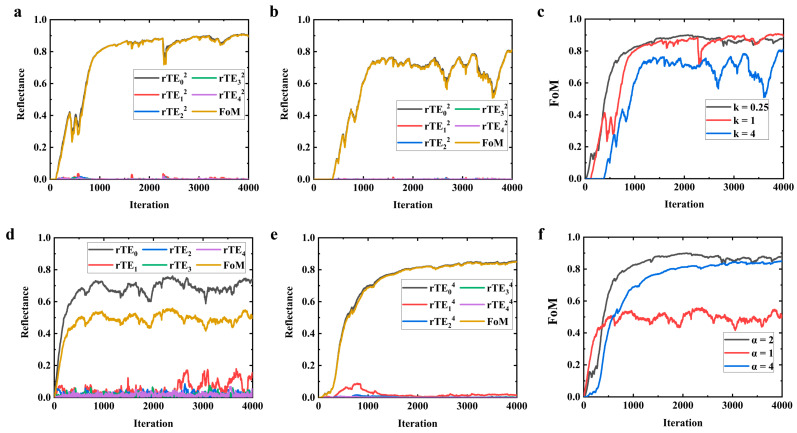
Comparison of convergence speeds and convergence characteristics for different *FoM*s: (**a**) *k* = 1; (**b**) *k* = 4; (**c**) comparison with benchmark (k = 0.25); (**d**) *α* = 1; (**e**) *α* = 4; (**f**) comparison with benchmark (*α* = 2).

**Figure 7 nanomaterials-14-00787-f007:**
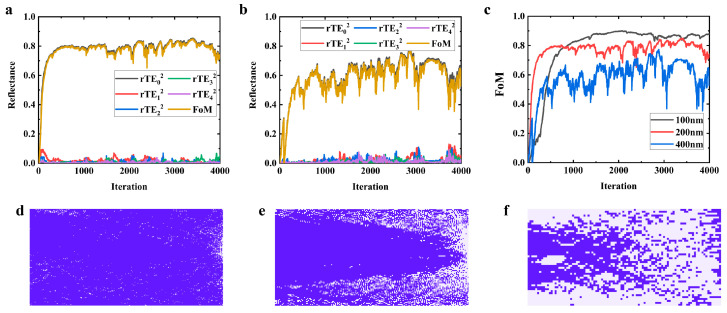
Simulation results for different etching sizes: (**a**) 200 nm pixel size; (**b**) 400 nm pixel size; (**c**) comparison with benchmark (100 nm pixel size). Optimized layout for different etching size: (**d**) 100 nm pixel size; (**e**) 200 nm pixel size; (**f**) 400 nm pixel size.

**Figure 8 nanomaterials-14-00787-f008:**
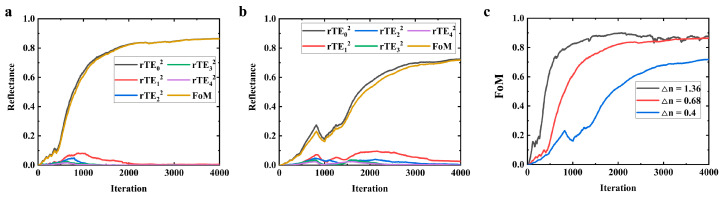
Simulation results for different refractive index differences (etching depths). (**a**) The difference of refractive index is 0.68; (**b**) The difference in refractive index is 0.4; (**c**) Comparison of optimization curves with different refractive index difference.

**Figure 9 nanomaterials-14-00787-f009:**
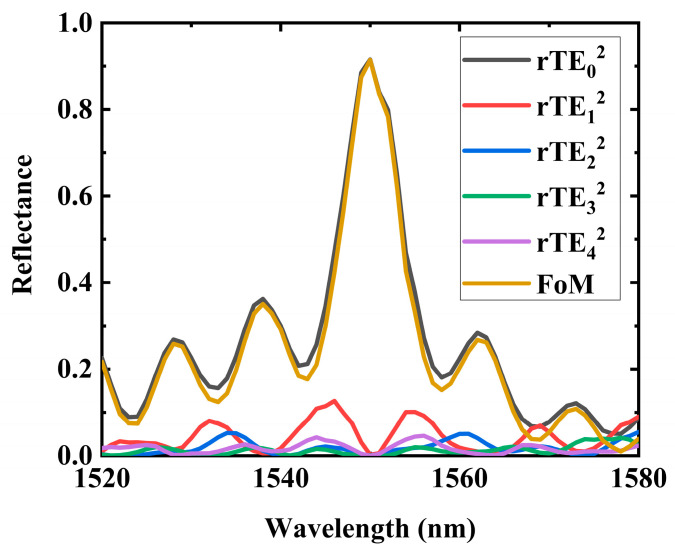
Operating bandwidth characteristic. In the operating wavelength range of 1549 to 1551 nm, the *FoM* exceeds 0.8.

**Figure 10 nanomaterials-14-00787-f010:**
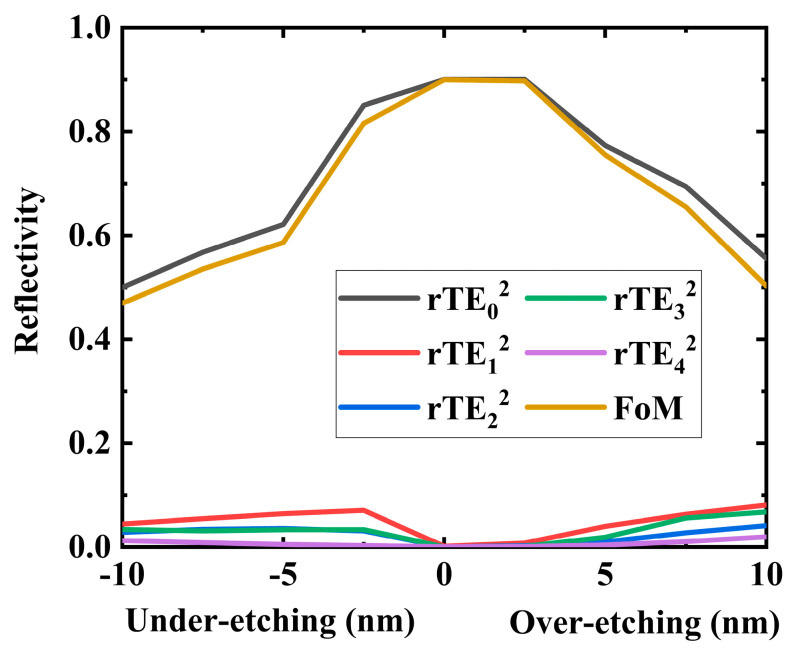
Under-etching and over-etching simulation.

**Table 1 nanomaterials-14-00787-t001:** Key parameters of BAL waveguide reflector.

Parameter Name	Value/Description
BAL Width	20 μm
Operation Wavelength	1550 nm
Background Refractive Index	1.46
Core Material Refractive Index	3.47
Mode Polarization	Transverse electric (TE)
Allowable Lateral Modes	5
Area of Design Region	40 × 20 μm^2^
Adjustable Pixel Size	0.1 × 0.1 μm^2^
Degree of Freedom	80,000
Optimization Algorithm	Inverse design using EIM and adjoint method

## Data Availability

The data presented in this study are available on the request from the corresponding author.

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
