# Peer review of "Adjoint Algorithm Design of Selective Mode Reflecting Metastructure for BAL Applications"

_nanomaterials, 2024, doi:10.3390/nano14090787_

Round 1

Reviewer 1 Report

Comments and Suggestions for Authors

Dear Authors and Editor!

The manuscript under considerations deals with an interesting and practically important problem, and thus it may be of substantial interest to the audience.

However, in my opinion the work can not be published in its present state due to very poor presentation quality. At least, Section 2 ("Design Principle and Optimization Results") should be completely re-written to provide solid description of investigated designs. It is unclear even what exactly is the "selective grating" and whether the calculation is 2D or 3D. The paper has to be extensively edited to become understandable, and so my recommendation is  to reject the current manuscript at this stage.

Comments on the Quality of English Language

Despite that minor issues are clearly visible in the Abstract, the quality of English is generally ok.

Reviewer 2 Report

Comments and Suggestions for Authors

It is an interesting idea but the authors should improve the presentation by addressing the following points:

1. Fig 3 is of too low quality - needs improvement. Similar improvement is needed for Figs, 5-9.

2. From the text and figures I cannot extract with certainty what is the width of the mirror. Is it 40 mi? The authors should state it clearly in many places because this is the main parameter that BAL design person is going to look for in this paper.

3. If the mirror with is 40 mi then it is a bit small for BAL. Can the authors try mirror widths of 200 mi or 400 mi? These are the widths that a BAL designer would wish to see. Is this approach applicable to such widths? Further, with 40 mi a designer would go for a tapered laser as it would most likely give sufficient filtering. Could the authors discuss how tapered laser compares with the proposed solution?

4. Is FD analysis 2D or 3D?

5. What is the 3D structure of the mirror?

6. Has a BAL with such mirror been fabricated?

Reviewer 3 Report

Comments and Suggestions for Authors

Li et al. developed an approach using digital adjoint sensitivity analysis to create a highly efficient mode-selective grating. The device features a footprint of 20×40 um2, which is unusually large in inverted design. After around five days of tuning, the structure attained a 90% reflection for the basic mode. It reduced the reflection rates of the first to fourth modes to below 0.2% at the same time, making it very appropriate as a reflecting surface for laser cavities. The work is interesting and well-written, however it needs small modifications before being published.

1.    The introduction lacks a clear explanation of the significance of the work and the challenges with the existing technique. The author could provide more points to emphasize the importance of the current research.

2.    The methods need more detail and a clearer explanation for better comprehension.

3.    The complexity of Figure 1 and 2 necessitates the author to include more comprehensible figures to enhance the clarity of their work.

4.    Figure 3-6 needs high resolutions.

5.    Inverse structure requires a schematic diagram for explanation.
